# BAIL: Best-Action Imitation Learning for Batch Deep Reinforcement Learning

**Xinyue Chen**[1]  **Zijian Zhou**[1]  **Zheng Wang**[1]

**Che Wang**[1,2]  **Yanqiu Wu**[1,2]  **Keith Ross**[1,2*]

[1] New York University Shanghai
[2] New York University

## Abstract

There has recently been a surge in research in batch Deep Reinforcement Learning (DRL), which aims for learning a high-performing policy from a given dataset without additional interactions with the environment. We propose a new algorithm, Best-Action Imitation Learning (BAIL), which strives for both simplicity and performance. BAIL learns a V function, uses the V function to select actions it believes to be high-performing, and then uses those actions to train a policy network using imitation learning. For the MuJoCo benchmark, we provide a comprehensive experimental study of BAIL, comparing its performance to four other batch Q-learning and imitation-learning schemes for a large variety of batch datasets. Our experiments show that BAIL's performance is much higher than the other schemes, and is also computationally much faster than the batch Q-learning schemes.

## 1  Introduction

The field of Deep Reinforcement Learning (DRL) has recently seen a surge in research in batch reinforcement learning, which is the problem of sample-efficient learning from a given dataset without additional interactions with the environment. Batch RL enables reusing the data collected by a policy to possibly improve the policy without further interactions with the environment, it has the potential to leverage existing large datasets to obtain much better sample efficiency. A batch RL algorithm can also be deployed as part of a growing-batch algorithm, where the batch algorithm seeks a high-performing exploitation policy using the data in an experience replay buffer [18], combines this policy with exploration to add fresh data to the buffer, and then repeats the whole process [15, 3]. Batch RL may also be necessary for learning a policy in safety-critical systems where a partially trained policy cannot be deployed online to collect data.

Fujimoto et al. [8] made the critical observation that when conventional Q-function based algorithms, such as Deep Deterministic Policy Gradient (DDPG), are directly applied to batch reinforcement learning, they learn very poorly, or even entirely diverge due to *extrapolation error*. Therefore, in order to obtain high-performing policies from batch data, new algorithms are required. Recent batch DRL algorithms roughly fall into two categories: Q-function-based algorithms such as BCQ [8] and BEAR [14]; and Imitation Learning (IL)-based algorithms such as MARWIL [27] and AWR [20].

We propose a new algorithm, Best-Action Imitation Learning (BAIL), which strives for both simplicity and performance. BAIL is an advanced IL method and its value estimates are updated only with data in the batch, giving stable estimates. BAIL not only provides state-of-the-art performance, it is also

computationally fast. Moreover, it is conceptually and algorithmically simple, thereby satisfying the principle of Occam's razor.

BAIL has three steps. In the first step, BAIL learns a V function by training a neural network to obtain the "upper envelope of the data". In the second step, it selects from the dataset the state action-pairs whose Monte Carlo returns are close to the upper envelope. In the last step, it simply trains a policy network with vanilla imitation learning using the selected actions. The method thus combines a novel approach for V-learning with IL.

Because the BCQ and BEAR codes are publicly available, we are able to make a careful and comprehensive comparison of the performance of BAIL, BCQ, BEAR, MARWIL and vanilla Behavior Cloning (BC) using the Mujoco benchmark. For our experiments, we create training batches in a manner identical to what was done in the BCQ paper (using DDPG [17] to create the batches), and add additional training batches for the environments Ant and Humanoid using SAC [10], giving a total of 22 training batches with non-expert data. Our experimental results show that BAIL wins for 20 of the 22 batches, with overall performance 42% or more higher than the other algorithms. Moreover, BAIL is computationally 30-50 times faster than the Q-learning algorithms. BAIL therefore achieves state-of-the-art performance while being significantly simpler and faster than BCQ and BEAR.

In summary, the contributions of this paper are as follows: $(i)$ BAIL, a new high-performing batch DRL algorithm, along with the novel concept of "the upper envelope of data"; $(ii)$ extensive, carefully-designed experiments comparing five batch DRL algorithms over diverse datasets. The computational results give significant insight into how different types of batch DRL algorithms perform for different types of data sets. We provide public open source code for reproducibility [2]. We will also make our datasets publicly available for future benchmarking.

## 2  Related work

Batch reinforcement learning in both the tabular and function approximator settings has long been studied [15, 21] and continues to be a highly active area of research [23, 13, 24, 4, 11, 12].

The difficulty of training deep neural networks effectively in the batch setting has been studied and discussed in a series of recent works, [16]. In the case of Q-function based methods, this difficulty is a combined result of extrapolation error and overestimation in Q updates [8, 6, 25].

Batch-Constrained deep Q-learning (BCQ) avoids the extrapolation error problem by constraining the set of actions over which the approximate Q-function is maximized [8]. More specifically, BCQ first trains a state-dependent Variational Auto Encoder (VAE) using the state action pairs in the batch data. When optimizing the approximate Q-function over actions, instead of optimizing over all actions, it optimizes over a subset of actions generated by the VAE. The BCQ algorithm is further complicated by introducing a perturbation model, which employs an additional neural network that outputs an adjustment to an action. BCQ additionally employs a modified version of clipped-Double Q-Learning to obtain satisfactory performance. Kumar et al. [14] recently proposed BEAR for batch DRL. Like BCQ, BEAR also constrains the actions over which it maximizes the approximate Q function. BEAR is relatively complex, employing Maximum Mean Discrepancy [9], kernel selection, a parametric model that fits a tanh-Gaussian distribution to the dataset, and a test policy that is different from the learned actor policy.

Monotonic Advantage Re-Weighted Imitation Learning (MARWIL) [27] uses exponentially weighted imitation learning, with the weights being determined by estimates of the advantage function. Advantage Weighted Regression (AWR) [20], another IL-based scheme, which is conceptually very similar to MARWIL, was primarily designed for online learning, but can also be employed in batch RL. BAIL, being an IL-based algorithm, shares some similarities with MARWIL and AWR; however, instead of weighting with advantage estimates, it uses the novel concept of the upper envelope to learn a V-function and select best actions, providing major performance improvements. In the online case, Self-Imitation Learning learns only from data that have a cumulative discounted return higher than the current value estimates to enhance exploration [19], this idea has also been explored in the multi-agent scenario [26].

Agarwal et al. [1] proposed Random Ensemble Mixture (REM), an ensembling scheme which enforces optimal Bellman consistency on random convex combinations of the Q-heads of a multi-headed Q-network. For the Atari 2600 games, batch REM can out-perform the policies used to collect the data. REM and BAIL are orthogonal, and it may be possible to combine them in the future to achieve even higher performance. Batch algorithms that apply to discrete actions space have been benchmarked on the Atari environments, showing that a discrete variant of BCQ has robust performance and outperform a number of other methods [7]. In the continuous action space case, MuJoCo has been the major benchmark. Very recently, other benchmarks environments have also been proposed for more systematic comparison and analysis [5].

## 3   Batch Deep Reinforcement Learning

We represent the environment with a Markov Decision Process (MDP) defined by a tuple $(\mathcal{S}, \mathcal{A}, g, r, \rho, \gamma)$, where $\mathcal{S}$ is the state space, $\mathcal{A}$ is the action space, $\rho$ is the initial state distribution, and $\gamma$ is the discount factor. The functions $g(s, a)$ and $r(s, a)$ represent the dynamics and reward function, respectively. In this paper we assume that the dynamics of the environment are deterministic, that is, there are real-valued functions $g(s, a)$ and $r(s, a)$ such that when in state $s$ and action $a$ is chosen, then the next state is $s' = g(s, a)$ and the reward received is $r(s, a)$. We note that all the simulated robotic locomotion environments in the MuJoCo benchmark are deterministic. Furthermore, many of the Atari game environments are deterministic [2]. Thus, from an applications perspective, the class of deterministic environments is a large and important class. Although we assume that the environment is deterministic, as is typically the case with reinforcement learning, we do not assume the functions $g(s, a)$ and $r(s, a)$ are known.

In batch reinforcement learning, we are provided a batch of $m$ data points $\mathcal{B} = \{(s_i, a_i, r_i, s'_i), i = 1, ..., m\}$. Using this batch, the goal is train a high-performing policy without any and further interaction with the environment. Typically the batch $\mathcal{B}$ is training data obtained while training a policy in some episodic fashion, or is execution data obtained with a fixed deterministic policy over multiple episodes. In the batch reinforcement learning problem, we do not have knowledge of the algorithm, policy, or seeds that were used to generate the episodes in the batch $\mathcal{B}$.

## 4   Best-Action Imitation Learning (BAIL)

In this paper we present BAIL, an algorithm that not only provides state-of-the-art performance on simulated robotic locomotion tasks, but is also fast and algorithmically simple. The motivation behind BAIL is as follows. For a given deterministic MDP, let $V^*(s)$ be the optimal value function. For a particular state-action pair $(s, a)$, let $G(s, a)$ denote a return using some policy when beginning state $s$ and choosing action $a$. Any action $a^*$ that satisfies $G(s, a^*) = V^*(s)$ is an optimal action for state $s$. Thus, ideally we would like to construct an algorithm which finds actions that satisfy $G(s, a^*) = V^*(s)$ for each state $s$.

In batch reinforcement learning, since we are only given limited data, we can only hope to obtain an approximation of $V^*(s)$. In BAIL, we first try to make the best possible estimate of $V^*(s)$ using only the limited information in the batch dataset. Call this estimate $V(s)$. We then select state-action pairs from the dataset whose associated returns $G(s, a)$ are close to $V(s)$. Finally, we train a policy with IL using the selected state-action pairs. Thus, BAIL combines both V-learning and IL. To obtain the estimate $V(s)$ of the value function, we introduce the "upper envelope of the data".

### 4.1   Upper envelope of the data

We first define a $\lambda$-regularized upper envelope, and then provide an algorithm for finding it. To the best of our knowledge, the notion of the upper envelope of a dataset is novel.

Recall that we have a batch of data $\mathcal{B} = \{(s_i, a_i, r_i, s'_i), i = 1, ..., m\}$. Although we do not assume we know what algorithm was used to generate the batch, we make the natural assumption that the data in the batch was generated in an episodic fashion, and that the data in the batch is ordered accordingly. For each data point $i \in \{1, \ldots, m\}$, we calculate the Monte Carlo return $G_i$ as the sum of the discounted rewards from state $s_i$ to the end of the episode as $G_i = \sum_{t=i}^{T} \gamma^{t-i} r_t$ where $T$ denotes the time at which the episode ends for the episode that contains the $i$th data point.

Having defined the return for each data point in the batch, we now seek an upper-envelope of the data $\mathcal{G} := \{(s_i, G_i), i = 1, ..., m\}$. Let $V_\phi(s)$ denote a neural networks parameterized by $\phi = (w, b)$ that takes as input a state $s$ and outputs a real number, where $w$ and $b$ denote the weights and bias, respectively. For a fixed $\lambda \geq 0$, we say that $V_{\phi^\lambda}(s)$ is a $\lambda$-regularized upper envelope for $\mathcal{G}$ if $\phi^\lambda$ is an optimal solution for the following constrained optimization problem:

$$\min_\phi \sum_{i=1}^m [V_\phi(s_i) - G_i]^2 + \lambda \|w\|^2 \qquad s.t. \qquad V_\phi(s_i) \geq G_i, \qquad i = 1, 2, \ldots, m \qquad (1)$$

Note that a $\lambda$-regularized upper envelope always lies above all the returns. The optimization problem strives to bring the envelope as close to the data as possible while maintaining regularization to prevent overfitting. The solution $\phi^\lambda$ to the constrained optimization problem may not be unique. Nevertheless, we have the following theorem to characterize the limiting behavior of $\lambda$-regularized upper envelopes.

**Theorem 4.1** *Suppose $V_\phi(s)$ is a multi-layer fully connected neural network with ReLu activation units, and there is a bias term at the output layer. For each $\lambda \geq 0$, let $V_{\phi^\lambda}(s)$ be a $\lambda$-regularized upper envelope for $\mathcal{G}$, that is, $\phi^\lambda = (w^\lambda, b^\lambda)$ is an optimal solution of the above constrained optimization problem. Then, we have*

*(1)* $\lim_{\lambda \to \infty} V_{\phi^\lambda}(s) = \max_{1 \leq i \leq m} \{G_i\}$ *for all $s \in \mathcal{S}$.*

*(2) When $\lambda = 0$, if there are sufficient number of activation units and layers, then $V_{\phi^0}(s)$ will interpolate the data in $\mathcal{G}$, i.e., $V_{\phi^0}(s_i) = G_i$ for all $i = 1, \ldots, m$.*

From the above theorem, we see that when $\lambda$ is very small, the upper envelope aims to interpolate the data, and when $\lambda$ is large, the upper envelope approaches a constant going through the data point with the highest return. Just as in classical regression, there is a sweet-spot for $\lambda$, the one that provides the best generalization.

We solve the constrained optimization problem (1) with a penalty-function approach. Specifically, to obtain an approximate upper envelope of the data $\mathcal{G}$, we solve an unconstrained optimization problem with a penalty loss function (with $\lambda$ fixed):

$$L^K(\phi) = \sum_{i=1}^m (V_\phi(s_i) - G_i)^2 \{\mathbb{1}_{(V_\phi(s_i) \geq G_i)} + K \cdot \mathbb{1}_{(V_\phi(s_i) < G_i)}\} + \lambda \|w\|^2 \qquad (2)$$

where $K >> 1$ is the penalty coefficient and $\mathbb{1}_{(.)}$ is the indicator function. For a finite $K$ value, the penalty loss function will produce an approximate upper envelope $V(s_i)$, since $V(s_i)$ may be slightly less than $G_i$ for some data points. In practice, we find $K = 1000$ works well for all environments tested. For $K \to \infty$, we have the following theoretical justification of the approximation:

**Theorem 4.2** *Let $\phi^K$ be a solution that minimizes $L^K(\phi)$ with penalty constant $K$. Let $\phi^*$ be a limit point of $\{\phi^K\}$. Then $V_{\phi^*}(s)$ is an exact $\lambda$-regularized upper envelope, i.e., $\phi^*$ is an optimal solution for the constrained optimization problem (1).*

In practice, instead of $L_2$ regularization, we employ a mechanism similar to early-stopping regularization. We split the data into a training set and validation set. During training, after every epoch, we check whether the validation error for the penalty loss function (2) decreases, and stop updating the network parameters when the validation loss increases repeatedly. We describe the details of the early-stopping scheme in the supplementary materials.

Figure 1 provides some examples of upper envelopes obtained with training sets consisting of 1 million data points. Each figure shows the upper envelope and the returns for one environment. To aid visualization, the states are ordered in terms of their upper envelope $V(s_i)$ values.

## 4.2 Selecting the best actions

BAIL employs the upper envelope to select the best $(s, a)$ pairs from the batch data $\mathcal{B}$. Let $V(s)$ denote the upper envelope obtained from minimizing the penalty loss function (2) for a fixed value of

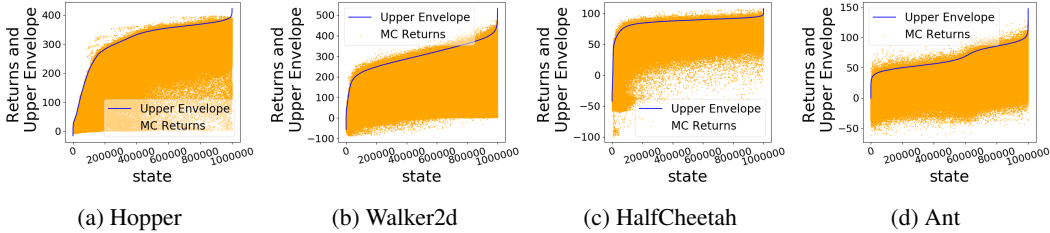

| (a) Hopper | (b) Walker2d | (c) HalfCheetah | (d) Ant |

Figure 1: Upper Envelopes trained on batches from different MuJoCo environments.

$K$. We consider two approaches for selecting the best actions. In the first approach, which we call BAIL-ratio, for a fixed $x > 0$, we choose all $(s_i, a_i)$ pairs from the batch data set $\mathcal{B}$ such that

$$G_i > xV(s_i) \tag{3}$$

We set $x$ such that $p\%$ of the data points are selected, where $p$ is a hyper-parameter. In this paper we use $p = 25\%$ for all environments and batches. In the second approach, which we call BAIL-difference, for a fixed $x > 0$, we choose all $(s_i, a_i)$ pairs from the batch data set $\mathcal{B}$ such that

$$G_i \geq V(s_i) - x \tag{4}$$

In our experiments, BAIL-ratio and BAIL-difference have similar performance, with BAIL-ratio sometimes a little better. We henceforth only consider BAIL-ratio, and simply refer to it as BAIL.

In summary, BAIL employs two neural networks. The first network is used to approximate the optimal value function based on the data in the batch $\mathcal{B}$. The second network is the policy, which is trained with imitation learning. We refer to the algorithm just described as BAIL. We also consider a variation, which we call Progressive BAIL, in which we train the upper envelope parameters $\phi$ and the policy network parameters $\theta$ in parallel rather than sequentially. Progressive BAIL doesn't change how we obtain the upper envelope, since the upper envelope does not depend on the policy parameters in either BAIL or Progressive BAIL. It does, however, affect the training of the policy parameters. We provide detailed pseudo-code for both BAIL and Progressive BAIL in the supplementary materials. Our experimental results show that BAIL and Progressive BAIL both perform well with about the same performance over all batches. But BAIL might be a better choice since it's much faster.

### 4.3 Augmented and oracle returns for the MuJoCo benchmark

Both BCQ and BEAR papers use the MuJoCo robotic locomotive benchmarks to gauge the performance of their algorithms [8] [14]. We will compare the performance of BAIL with BCQ, BEAR, MARWIL and BC using the same MuJoCo environments.

The MuJoCo environments are naturally infinite-horizon non-episodic continuing-task environments [22]. During training, however, researchers typically create artificial episodes of maximum length 1000 time steps; after 1000 time steps, a random initial state is chosen and a new episode begins. This means that to apply BAIL, we need to approximate infinite-horizon discounted returns using the finite-length episodes in the data set. For data points appearing near the beginning of the episode, the finite-horizon return will closely approximate the (idealized) infinite-horizon return due to discounting; but for a data point near the end of an episode, the finite horizon return can be inaccurate and should be augmented. To calculate the augmentation for the $i$th data point, we use the following heuristic. Let $\mathcal{E} \subset \mathcal{B}$ denote the episode of data that contains the $i$th data point, and let $s'$ be the last state in episode $\mathcal{E}$. We then set $s_j$ to be the state in the first $\max\{1000 - i, 200\}$ data points of the episode $\mathcal{E}$ that is closest (in Euclidean norm) to the "terminal state" $s'$. We then set

$$G_i = \sum_{t=i}^{T} \gamma^{t-i} r_t + \gamma^{T-i+1} \sum_{t=j}^{T} \gamma^{t-j} r_t \tag{5}$$

Note that $G_i$ in (5) will have at least 800 terms, so there is no need for additional terms due to the discounting. Importantly, the rewards in the two sums in (5) are generated by the same policy. The first sum uses the actual rewards accrued until to the end of the episode; the second sum approximates what the actual rewards would have been if the episode was allowed to continue past 1000 time steps.

Table 1: Performance of five Batch DRL algorithms for 22 different training datasets.

| ENVIRONMENT | BAIL | BCQ | BEAR | BC | MARWIL |
|---|---|---|---|---|---|
| $\sigma = 0.1$ HOPPER B1 | **2173 ± 291** | 1219 ± 114 | 505 ± 285 | 626 ± 112 | 827 ± 220 |
| $\sigma = 0.1$ HOPPER B2 | **2078 ± 180** | 1178 ± 87 | 985 ± 3 | 579 ± 141 | 620 ± 336 |
| $\sigma = 0.1$ WALKER B1 | **1125 ± 113** | 576 ± 309 | 610 ± 212 | 514 ± 17 | 436 ± 24 |
| $\sigma = 0.1$ WALKER B2 | **3141 ± 300** | 2338 ± 388 | 2707 ± 425 | 1741 ± 239 | 1810 ± 200 |
| $\sigma = 0.1$ HC B1 | **5746 ± 29** | **5883 ± 43** | 0 ± 0 | **5546 ± 29** | **5573 ± 35** |
| $\sigma = 0.1$ HC B2 | 7212 ± 43 | **7562 ± 31** | 0 ± 0 | 6765 ± 108 | 6828 ± 111 |
| $\sigma = 0.5$ HOPPER B1 | **2054 ± 158** | 1145 ± 300 | 203 ± 42 | 919 ± 52 | 946 ± 103 |
| $\sigma = 0.5$ HOPPER B2 | **2623 ± 282** | 1823 ± 555 | 241 ± 239 | 694 ± 64 | 818 ± 112 |
| $\sigma = 0.5$ WALKER B1 | **2522 ± 51** | 1552 ± 455 | 1248 ± 181 | 2178 ± 178 | 2111 ± 52 |
| $\sigma = 0.5$ WALKER B2 | **3115 ± 133** | 2785 ± 123 | 2302 ± 630 | 2483 ± 94 | 2364 ± 228 |
| $\sigma = 0.5$ HC B1 | 1055 ± 9 | **1222 ± 38** | 924 ± 579 | 570 ± 35 | 512 ± 43 |
| $\sigma = 0.5$ HC B2 | 7173 ± 120 | 5807 ± 249 | −114 ± 140 | **6545 ± 171** | **6668 ± 93** |
| SAC HOPPER B1 | **3296 ± 105** | 2681 ± 438 | 1000 ± 110 | 2853 ± 318 | 2897 ± 227 |
| SAC HOPPER B2 | 1831 ± 915 | **2134 ± 917** | 1139 ± 317 | **2240 ± 367** | 2063 ± 168 |
| SAC WALKER B1 | **2455 ± 211** | **2408 ± 84** | −3 ± 5 | 1674 ± 277 | 1484 ± 140 |
| SAC WALKER B2 | **4767 ± 130** | 3794 ± 398 | 325 ± 75 | 2599 ± 145 | 2651 ± 268 |
| SAC HC B1 | **10143 ± 77** | 8607 ± 473 | 7392 ± 257 | 8874 ± 221 | 9105 ± 90 |
| SAC HC B2 | **10772 ± 59** | **10106 ± 134** | 7217 ± 273 | 9523 ± 164 | 9488 ± 136 |
| SAC ANT B1 | **4284 ± 64** | **4042 ± 113** | 3452 ± 128 | 3986 ± 112 | 4033 ± 130 |
| SAC ANT B2 | **4946 ± 148** | **4640 ± 76** | 3712 ± 236 | 4618 ± 111 | 4589 ± 130 |
| SAC HUMANOID B1 | **3852 ± 430** | 1411 ± 250 | 0 ± 0 | 543 ± 378 | 589 ± 121 |
| SAC HUMANOID B2 | **3565 ± 153** | 1221 ± 207 | 0 ± 0 | 1216 ± 826 | 1033 ± 257 |

To validate this heuristic, we compute oracle returns by letting episodes run up to 2000 time steps. In this manner, every return is calculated with at least 1000 actual rewards, and is therefore essentially exact due to discounting. With oracle returns, we can analyze the effect of the augmentation heuristic. Specifically, we compared the performance of BAIL with the augmentation heuristic and with oracle for Hopper-v2 for seven diverse batches. Learning curves are shown in the supplementary materials. For all five batches, the augmentation heuristic has similar performance compared to the oracle. We conclude that our augmentation heuristic is a satisfactory method for addressing continual environments such as MuJoCo, which is also confirmed with good performance in Table 1 .

# 5 Experimental results

Along with the BAIL algorithm, the experimental results are the main contribution of this paper. Using 62 diverse batches (many of which are similar to those used in the BCQ and BEAR papers), we provide a comprehensive comparison of five algorithms: BAIL, BCQ, BEAR, MARWIL and BC. We use authors' code and recommended hyper-parameters when available, and we strive to make the comparisons as fair as possible.

## 5.1 Batch generation

We provide experimental results for five MuJoCo environments: HalfCheetah-v2, Hopper-v2, Walker2d-v2, Ant-v2, and Humanoid-v2. To make the comparison as favorable as possible for BCQ and BEAR, we generate datasets using the same procedures proposed in the BCQ and BEAR papers, and compare the algorithms using those data sets. The BCQ [8] and BEAR [14] papers both use batch datasets of one million samples, but generate the batches using different approaches. One important observation we make, which was not brought to light in previous batch DRL papers, is that batches generated with different seeds but with otherwise exactly the same algorithm can give drastically different results for batch DRL. Because of this, for every experimental scenario considered in this paper, we generate two batches, each generated with a different random seed.

### 5.1.1 Training batches

We generate batches *while* training DDPG [17] from scratch with exploration noise of $\sigma = 0.5$ for HalfCheetah-v2, Hopper-v2, and Walker2d-v2, as exactly done in the BCQ paper. We also generate batches with $\sigma = 0.1$ to study the robustness of tested algorithms with lower noise level. We also generate training batches for all five environments by training policies with adaptive Soft Actor Critic (SAC) [10]. This gives six DDPG and five SAC scenarios. For each one, we generate two batches with different random seeds, giving a total of 22 "training batches" composed of non-expert data. These batches are the most important ones to measure the performance of a batch method, since they contain sub-optimal data obtained from the training process well before optimal performance is achieved (and in many cases using sub-optimal algorithms for training), which is difficult for vanilla behavioral cloning to use.

### 5.1.2 Execution batches

In addition to training batches, we also study execution batches. We do a similar procedure as in the BEAR paper: first train SAC [10] for a certain number of environment interactions, then *fixes the trained policy* and generates one million "execution data points". The BEAR paper generates batches with "mediocre data" where training is up to a mediocre performance, and with "optimal data" where training is up to near optimal performance. When generating the batches with the trained policy, the BEAR paper continues to include exploration noise, using the trained $\sigma(s)$ values in the policy network. Since after training, a test policy is typically deployed without exploration noise, we also consider noise-free data generation. The BEAR paper considers the same five MuJoCo environments considered here. This gives rise to 20 scenarios. For each one we generate two batches with random seeds, giving a total of 40 "execution batches".

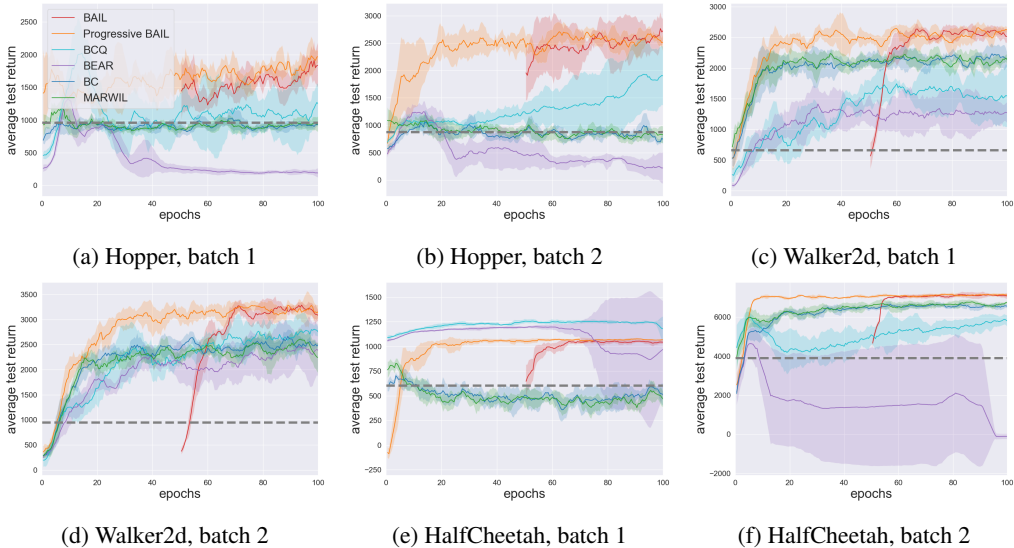

(a) Hopper, batch 1      (b) Hopper, batch 2      (c) Walker2d, batch 1

(d) Walker2d, batch 2      (e) HalfCheetah, batch 1      (f) HalfCheetah, batch 2

Figure 2: Learning curves using DDPG training batches with $\sigma = 0.5$.

### 5.2 Performance comparison

We now carefully compare the five algorithms. For a fair comparison, we keep all hyper-parameters fixed for all experiments, instead of fine-tuning for each one. For BCQ we use the authors' code with their default hyper-parameters. For BEAR we use the authors' code with their version "0" with "use ensemble variance" set to False and employ the recommended hyper-parameters. Because the MARWIL code is not publicly available, we write our own code, and use neural networks the same size as in BAIL. In the supplementary material we provide more details on implementations, and explain how the comparisons are carefully and fairly done.

For each algorithm, we train for 100 epochs (with each epoch consisting of one million data points). For each algorithm, after every 0.5 epochs, we run ten test episodes with the current policy to evaluate

performance. We then repeat the procedure for five seeds to obtain the mean and confidence intervals shown in the learning curves. Due to page length limitations, we cannot present the learning curves for all 62 datasets. Here we focus on the 6 DDPG training data batches with $\sigma = 0.5$ (corresponding to the datasets in the BCQ paper), and present the learning curves for the other batches in the supplementary material. However, we present summary results for all datasets in this section.

Figure 2 shows the learning curves for the 6 DDPG data sets over 100 epochs. As is commonly done, we present smoothed average performance and standard deviations. Note that for BAIL, all curves start at 50 epochs. This provides a fair comparison, since for BAIL we first use 50 epochs of data to train the upper envelopes and then use imitation learning to train the policy network. The horizontal grey dashed line indicates the average return of episodes contained in the batch.

Table 1 presents our main results, comparing the five algorithms for the 22 training batches. In batch DRL, since there is no interaction with the environment, one cannot use the policy parameters that provided the highest test returns to assess performance. So in Table 1 , for each algorithm, we assume that the practitioner would use the policy obtained after 100 epochs of training. Since there can be significant variation in performance from one policy to the next during training, we calculate the average performance across epochs 95.5 to 100 (i.e., averaged over the last ten tested policies). We do this for each of the five training seeds. We then report the average and standard deviation of these values across the five seeds. For each batch, all the algorithms that are within 10% of the highest average value are considered winners and are indicated in bold.

From Table 1 we observe that for the training batches, BAIL is the clear winner. BAIL wins for 20 of the 22 batches, and BCQ is in second place winning for only 8 of the 22 batches. These results show that BAIL is very robust for a wide variety of training datasets, including non-expert datasets, datasets generated as described in the BCQ paper, and for datasets with the more challenging Ant and Humanoid environments. To evaluate the average performance improvement of BAIL over BCQ, for each batch we take the ratio of the BAIL performance to the BCQ performance and then average over the 22 bathes. We also do the same for BC. With this metric, BAIL performs 42% better than BCQ, and 101% better than BC. BAIL is also more stable across seeds: The normalized standard deviations (standard deviation divided by average performance) of BAIL, averaged over the 22 batches, is about half that of BCQ. Because BAIL performs so well for training batches, BAIL can potentially be successfully used for growing batch DRL.

We also note that BEAR occasionally performs very poorly. This is likely because we are using one set of recommended BEAR hyper-parameters for all environments, whereas the BEAR paper reports results using different hyper-parameters for different environments. We also note that for the MuJoCo environments, MARWIL performs similarly to BC.

For the execution batches, the results are given in the supplementary materials. When BAIL uses the same hyper-parameters as for training batches (though fine-tuning will yield better results, we strive for a fair comparison), BC, MARWIL, BAIL, and BCQ have similar overall performance, with BC being the most robust and the overall winner. Comparing BAIL and BCQ, BAIL has slightly stronger average performance score, and BCQ has a few more wins. It is no surprise that BC is the strongest here, since the execution batches are generated with a single fixed policy and are easy for BC to learn well. These results imply that the focus of future research on batch DRL should be on training batches, or other diverse datasets, since vanilla BC already works very well for fixed-policy datasets.

BAIL uses an upper envelope to select the "best" data points for training a policy network with imitation learning. It is natural to ask how BAIL would perform when using the more naive approach of selecting the best actions by simply selecting the same percentage of data points with the highest $G_i$ values, and also by constructing the value function with regression rather than with an upper envelope. These schemes do not do as well as BAIL by a wide margin (see supplementary material).

Intuitively, BAIL can perform better than BCQ and BEAR because these policy-constraint methods rely on carefully tuned constraints to prevent the use of out-of-distribution actions. A loose constraint can cause extrapolation error to accumulate quickly, and a tight constraint will prevent the policy from choosing some of the good actions. BAIL, however, identifies and imitates the highest-performing actions in the dataset, thus avoiding the need to carefully tune such a constraint.

### 5.3 Comparison of run times for Batch DRL algorithms

In our experiments we run all algorithms each for $100$ epochs for five seeds for each batch. For training with one seed, it takes $1 \sim 2$ hours for BAIL (including the time for upper envelope training and imitation learning), $12 \sim 24$ hours for Progressive BAIL, $36 \sim 72$ hours for BCQ and $60 \sim 100$ hours for BEAR on a CPU node. Thus, roughly speaking, training BAIL is roughly 35 times faster than BCQ and 50 times faster than BEAR.

## 6 Conclusion

In conclusion, our experimental results show that $(i)$ for the training data batches, BAIL is the clear winner, winning for 20 of 22 batches with a performance improvement of 42% over BCQ and 101% over BC; $(ii)$ for the execution batches, vanilla BC does well with not much room for improvement, although BAIL and BCQ are almost as good and occasionally beat BC by a small amount; $(iii)$ BAIL is computationally much faster than the Q-learning-based algorithms BCQ and BEAR.

The results in this paper show that it is possible to achieve state-of-the art performance with a simple, computationally fast IL-based algorithm. BAIL is based on the notion of the "upper envelope of the data", which appears to be novel and may find applications in other machine-learning domains. One potential future research direction is to combine batch methods such as BAIL with exploration techniques to build robust online algorithms for better sample efficiency. Another potential direction is to develop methods that are more robust across different batches and hyperparameters and study what makes them robust. Such robustness can greatly improve computation time, and might be safer to work with when deployed to real-world systems.

## Broader Impact

This research may potentially lead to mechanisms for training robots and self-driving vehicles to perform complex tasks. Batch RL enables reusing the data collected by a policy to possibly improve the policy without further interactions with the environment. A batch RL algorithm can be deployed as part of a growing-batch algorithm, where the batch algorithm seeks a high-performing exploitation policy using the data in an experience replay buffer, combines this policy with exploration to add fresh data to the buffer, and then repeats the whole process. Batch RL may also be necessary for learning a policy in safety-critical systems where a partially trained policy cannot be deployed online to collect data. Compared to policy constraint methods discussed in the paper, BAIL uses a much smaller amount of computation to achieve good performance, its efficiency means a smaller computation cost and less consumption of energy.

## Acknowledgments and Disclosure of Funding

This research was partially supported by Nokia Bell Labs.

## Footnotes

*Correspondence to: Keith Ross <keithwross@nyu.edu>.

[2] https://github.com/lanyavik/BAIL

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
