[Supplementary Material]

# Supplementary Material for
# BAIL: Best-Action Imitation Learning for Batch Deep Reinforcement Learning

## A Proofs of Theorems

### A.1 Proof of Theorem 4.1

*Proof.* Part (1): For any $\lambda \geq 0$ and $\phi = (w, b)$ define

$$J^\lambda(\phi) = \sum_{i=1}^m [V_\phi(s_i) - G_i]^2 + \lambda\|w\|^2 \tag{1}$$

Note that for any $\phi$ of the form $\phi = (0, b)$, we have $V_{(0,b)}(s) = b$ for all $s$ and $J^\lambda(0, b) = \sum_{i=1}^m (b - G_i)^2$ for all $\lambda \geq 0$. Also define

$$G^* := \max_{1 \leq i \leq m} \{G_i\}$$

and $\hat{\phi} = (0, G^*)$. Note that $\hat{\phi}$ is feasible for the constrained optimization problem. It therefore follows that for any $\lambda \geq 0$:

$$J^\lambda(\phi^\lambda) \leq J^\lambda(\hat{\phi}) = \sum_{i=1}^m (G^* - G_i)^2 := H^* \tag{2}$$

We first show that $\lim_{\lambda \to \infty} w^\lambda = 0$. To proceed with a proof by contradiction, assume that this is not true. There then exists an $\epsilon > 0$ such that for any $\lambda \geq 0$ there exists some $\lambda' \geq \lambda$ such that $\|w^{\lambda'}\|^2 > \epsilon$. Choosing $\lambda = H^*/\epsilon$, we have for some $\lambda'$:

$$J^{\lambda'}(\phi^{\lambda'}) \geq \lambda'\|w^{\lambda'}\|^2 > \lambda \cdot \epsilon = H^* \tag{3}$$

But this contradicts (2), establishing $\lim_{\lambda \to \infty} w^\lambda = 0$.

Next, we show $\lim_{\lambda \to \infty} b^\lambda = G^*$. To prove this, we will show $\bar{b} := \limsup_{\lambda \to \infty} b^\lambda = G^*$ and also $\tilde{b} := \liminf_{\lambda \to \infty} b^\lambda = G^*$. First, consider a subsequence $\{b^{\lambda_n}\}$ such that $\lim_{n \to \infty} b^{\lambda_n} = \tilde{b}$. Due to the continuity of $\phi \to V_\phi(s)$ and $\lim_{\lambda \to \infty} w^\lambda = 0$, we have

$$\lim_{n \to \infty} V_{\phi^{\lambda_n}}(s) = V_{(0,\tilde{b})}(s) = \tilde{b} \qquad \forall s \tag{4}$$

Moreover, since $\phi^{\lambda^n}$ has to satisfy the constraints, we also have

$$V_{\phi^{\lambda_n}}(s_j) \geq G_j = G^*, \qquad i = 1, \ldots, m \tag{5}$$

where $j = \arg\max_i G_i$. Therefore, combining (4) and (5) yields

$$\liminf_{\lambda \to \infty} b^\lambda \geq G^* \tag{6}$$

Similarly, consider another subsequence $\{b^{\lambda_k}\}$ such that $\lim\limits_{k\to\infty} b^{\lambda_k} = \bar{b}$. Again, we have

$$\lim_{k\to\infty} V_{\phi^{\lambda_k}}(s) = \bar{b} \qquad \forall s \tag{7}$$

We have from (2) that

$$J^{\lambda_k}(\phi^{\lambda_k}) \le H^* \tag{8}$$

Letting $k \to \infty$ gives

$$\sum_{i=1}^{m}(\bar{b} - G_i)^2 \le \sum_{i=1}^{m}(G^* - G_i)^2 \tag{9}$$

which implies that

$$\bar{b} = \limsup_{\lambda\to\infty} b^\lambda \le G^* \tag{10}$$

Therefore, combining (6) and (10) together, we have finally shown that $\lim\limits_{\lambda\to\infty} b^\lambda = G^*$. As we have previously shown $\lim\limits_{\lambda\to\infty} w^\lambda = 0$, it follows that

$$\lim_{\lambda\to\infty} V_{\phi^\lambda}(s) = \max_{1\le i\le m}\{G_i\}, \qquad \forall s \tag{11}$$

Part (2): For the case of $\lambda = 0$, notice that we have finitely many inputs $s_i$ to feed into the neural network. Therefore, this is a typical problem regarding the *finite-sample expressivity* of neural networks, and the proof directly follows from the work done in [14]. $\square$

### A.2  Proof of Theorem 4.2

*Proof.* Let $J^\lambda(\phi) = \sum_{i=1}^{m}[V_\phi(s_i) - G_i]^2 + \lambda\|w\|^2$ be the loss function that defines the $\lambda$-regularized upper envelope. We notice that the penalty loss function $L^K(\phi)$ takes the form:

$$L^K(\phi) = J^\lambda(\phi) + (K-1)\sum_{i=1}^{m}(\max\{0, G_i - V_\phi(s_i)\})^2 \tag{12}$$

The theorem then directly follows from the standard convergence theorem for penalty functions [7]. $\square$

# B  Algorithmic Implementation

## B.1  Pseudo-Code and Early Stopping Scheme for Upper Envelope Training

BAIL includes a regularization scheme to prevent over-fitting when generating the upper envelope. We refer to it as an "early stopping scheme" because the key idea is to return to the parameter values which gave the lowest validation error (see Section 7.8 of Goodfellow et al. [3]). In our implementation, we initialize two upper envelope networks with parameters $\phi$ and $\phi'$, where $\phi$ is trained using the penalty loss, and $\phi'$ records the parameters with the lowest validation error encountered so far. The procedure is done as follows: After every epoch, we calculate the validation loss $L_\phi$ as the penalty loss over all the data in the validation set $\mathcal{B}_v$. We compare this validation loss $L_\phi$ to $L_{\phi'}$, which is the minimum validation loss encountered so far (throughout the history of training). If $L_\phi < L_{\phi'}$, we set $\phi' \leftarrow \phi$. If $L_\phi > L_{\phi'}$, we count the number of consecutive times this occurs. The training parameters $\phi$ are returned to $\phi'$ once there are $C$ consecutive times with $L_\phi > L_{\phi'}$. We use $C = 4$ in practice.

---

**Algorithm 1** BAIL

---

Initialize upper envelope parameters $\phi, \phi'$, policy parameters $\theta$. Obtain batch data $\mathcal{B}$. Randomly split data into training set $\mathcal{B}_t$ and validation set $\mathcal{B}_v$ for the upper envelope.
Compute return $G_i$ for each data point $i$ in $\mathcal{B}$.
Obtain upper envelope by minimizing the loss $L^K(\phi)$:
**for** $j = 1, \ldots, J$ **do**
    Sample a mini-batch $B$ from $\mathcal{B}$.
    Update $\phi$ using the gradient: $\nabla_\phi \sum_{i \in B}(V_\phi(s_i) - G_i)^2 \{\mathbb{1}_{(V_\phi(s_i) > G_i)} + K\mathbb{1}_{(V_\phi(s_i) < G_i)}\} + \lambda\|\phi\|^2$

    **if** time to do validation for the upper envelope **then**
        Compute validation loss on $B_v$
        Update $\phi$ and $\phi'$ according to the validation loss
    **end if**
**end for**
Select data point $i$ if $G_i > xV_\phi(s_i)$, where $x$ is such that $p\%$ of data in $\mathcal{B}$ are selected. Let $\mathcal{U}$ be the set of selected data points.
**for** $l = 1, \ldots, L$ **do**
    Sample a mini-batch $U$ of data from $\mathcal{U}$.
    Update $\theta$ using the gradient: $\nabla_\theta \sum_{i \in U}(\pi_\theta(s_i) - a_i)^2$
**end for**

---

**Algorithm 2** Progressive BAIL

---

Initialize upper envelope parameters $\phi, \phi'$, policy parameters $\theta$.
Obtain batch data $\mathcal{B}$. Randomly split data into training set $\mathcal{B}_t$ and validation set $\mathcal{B}_v$ for the upper envelope.
Compute return $G_i$ for each data point $i$ in $\mathcal{B}$.
**for** $l = 1, \ldots, L$ **do**
    Sample a mini-batch of data $B$ from the batch $\mathcal{B}_t$.
    Update $\phi$ using the gradient: $\nabla_\phi \sum_{i \in B_t}(V_\phi(s_i) - G_i)^2 \{\mathbb{1}_{(V_\phi(s_i) > G_i)} + K\mathbb{1}_{(V_\phi(s_i) < G_i)}\} + \lambda\|\phi\|^2$
    **if** time to validate **then**
        Compute validation loss on $B_v$
        Update $\phi$ and $\phi'$ according to validation loss
    **end if**
    Select data point $i$ if $G_i > xV_\phi(s_i)$, where $x$ is such that $p\%$ of data in $B$ are selected. Let $U$ be the set of selected data points.
    Update $\theta$ using the gradient: $\nabla_\theta \sum_{i \in U}(\pi_\theta(s_i) - a_i)^2$
**end for**

---

The pseudo-code for BAIL and Progressive BAIL, which include the early stopping scheme, are presented in Algorithms 1 and 2. Note BAIL has two for loops in series, whereas Progressive BAIL has only one for loop.

## B.2 Hyper-parameters of BAIL

BAIL and Progressive BAIL use the same hyper-parameters except for the selection percentage $p$. Details are provided in Table 1.

Table 1: BAIL hyper-parameters

| Parameter | Value |
|---|---|
| discount rate $\gamma$ | 0.99 |
| horizon $T$ | 1000 |
| training set size | $0.8 \cdot |\mathcal{B}|$ |
| validation set size | $0.2 \cdot |\mathcal{B}|$ |
| optimizer | Adam [4] |
| percentage $p\%$ | 30% for BAIL |
| | 25% for Progressive BAIL |
| **upper envelope network** | |
| structure | $128 \times 128$ hidden units, ReLU activation |
| learning rate | $3 \cdot 10^{-3}$ |
| penalty loss coefficient $K$ | 1000 |
| **policy network** | |
| structure | $400 \times 300$ hidden units, ReLU activation |
| learning rate | $1 \cdot 10^{-3}$ |

## C Experimental Details

This paper compares BAIL (our algorithm) with four other baselines: BC, BCQ, BEAR, and MARWIL. We use five MuJoCo environments, including Humanoid, which is the most challenging of the MuJoCo environments, and is not attempted in most other papers on batch DRL.

### C.1 Hyper-parameter consistency

When designing RL algorithms, it is desirable that they generalize over new, unforeseen environments and tasks. Therefore, consistent with common practice for online reinforcement learning [8, 9, 10, 6, 1, 11], when evaluating any given algorithm, we use the same hyper-parameters for all environments and all batches. The BCQ paper [2] also uses the same hyper-parameters for all experiments.

Alternatively, one could optimize the hyper-parameters for each environment separately. Not only is this not standard practice, but to make a fair comparison across all algorithms, this would require, for *each* of the five algorithms, performing a separate hyper-parameter search for *each* of the five environments.

### C.2 Reproduction of the Baseline Algorithms

In our submission, we went the extra mile to make a fair comparison to other batch RL algorithms. We are therefore confident about properly using the authors' BCQ and BEAR code, and fairly reproducing MARWIL for the MuJocO benchmark.

**BCQ** We use the authors' code and recommended hyper-parameters. In the BCQ paper, the "final buffer" batches are where the BCQ algorithm shines the most; therefore, included in our training batches are batches for which we used exactly the same "final buffer" experimental set-up. In our terminology, this corresponds to DDPG training batches with sigma = 0.5. Looking at the BCQ final-buffer results in Figure 2 and Table 1, we see that they are consistent with the results in Figure 2a in the BCQ paper.

**BEAR** To ensure that we are running the BEAR code properly, we obtained a dataset directly from the BEAR authors and ran the BEAR algorithm with a specific set of hyper-parameters among their recommendations. Specifically, we used their version "0" with "use ensemble variance" set to False and employ Laplacian kernel. The dataset provided by the authors was for Hopper-v2 with mediocre performance. The performance we obtained is shown in Figure 1, which fully matches the Hopper-v2 case in Figure 3 in [5]. Also, we observed that for some of our batches, we obtained very similar results to what is shown in the BEAR paper.

Figure 1: Our results when we apply BEAR to the authors' dataset. This figure matches Figure 3 in Kumar et al. [5].

However, the results shown in Tables 1 and 2 show that BEAR can sometimes have poor performance, much worse than what is shown in [5, 13]. This is because in [5, 13], hyper-parameters are optimized separately for each of the MuJoCo environments. In this paper, as discussed above, for each algorithm we use one set of hyper-parameters. In the case of BEAR, we use one of their recommended hyper-parameter settings for all environments and batches, namely, their version "0" with "use ensemble variance" set to False and employ the Laplacian kernel.

**MARWIL** The authors of MARWIL do not provide an open-source implementation of their algorithm. Furthermore, experiments in [12] are carried out on environments like HFO and TORCS which are considerably different from MuJoCo. We replicate all implementation details discussed in MARWIL, except that we use the same network architectures used for BCQ, BEAR and BAIL to ensure a fair comparison. We use the same augmentation heuristic for the returns as we use in BAIL. We use the recommended hyper-parameters given by the MARWIL authors.

## C.3 Common Hyper-parameters across all batch RL algorithms

**Network size** A common feature among all the batch DRL algorithms is that they have a policy neural network. BCQ and BEAR both have an architecture consisting of $400 \times 300$ hidden units with ReLU activation units. We use exactly the same network architecture for the policy network for BAIL and Progressive BAIL. For the IL-based algorithms, we also use this same policy network architecture.

**Learning rate** All algorithms considered in our experiments use the same learning rate of $1 \cdot 10^{-3}$ for the policy network, which is also the default in BCQ and BEAR.

## C.4 Evaluation methodology employed for all batch RL algorithms

To evaluate the performance of the current policy during training, we run ten episodes of test runs with the current policy and record the average of the returns. This is done with the same frequency for each algorithm considered in our experiments.

For a test episode, we sometimes encounter an error signal from the MuJoCo environment, and thus are not able to continue the episode. In these cases, we assign a zero value to the return for the terminated episode. In Tables 1 and 2 of the paper, there are a few entries with zero mean and zero standard deviation. These zeros are due to repeatedly encountering this error signal for the test runs using different seeds, with each test run getting a zero value for the return. This happens for BEAR in several batches, which is likely because we are not using different hyper-parameters for each environment.

# D   Ablation studies for BAIL

## D.1   Augmented return versus oracle performance

To validate our heuristic for the augmented returns, we compute oracle returns by letting episodes run up to 2000 time steps. In this manner, every return is calculated with at least 1000 actual rewards, and is therefore essentially exact due to discounting. Figure 2 compares the performance of BAIL using our augmentation heuristic and BAIL using the oracle for Hopper-v2 for seven diverse batches. The results show that our augmentation heuristic typically achieves oracle-level performance. We conclude that our augmentation heuristic is a satisfactory method for addressing continual environments such as MuJoCo, which is also confirmed with its good performance shown in Table 2 .

(a) DDPG training batch with $\sigma = 0.5$

(b) DDPG training batch with $\sigma = 0.1$

(c) SAC mediocre execution batch with $\sigma = 0$

(d) SAC mediocre execution batch with $\sigma = \sigma(s)$

(e) SAC optimal execution batch with $\sigma = 0$

(f) SAC optimal execution batch with $\sigma = \sigma(s)$

(g) SAC training batch

Figure 2: Augmented Returns versus Oracle Performance. All learning curves are for the Hopper-v2 environment. The x-axis ranges from 50 to 100 epochs since this comparison involves only BAIL. The results show that the augmentation heuristic typically achieves oracle-level performance.

## D.2 Ablation study for data selection

BAIL uses an upper envelope to select the "best" data points for training a policy network with imitation learning. It is natural to ask how BAIL would perform when using the more naive approach of selecting the best actions by simply selecting the same percentage of data points with the highest $G_i$ values. Figure 3 compares BAIL with the algorithm that simply chooses the state-action pairs with the highest returns (without using an upper envelope). The learning curves show that the upper envelope is a critical component of BAIL.

(a) Hopper $\sigma = 0.5$ 1st    (b) Hopper $\sigma = 0.5$ 2nd    (c) Hopper $\sigma = 0.1$ 1st    (d) Hopper $\sigma = 0.1$ 2nd

(e) Walker2d $\sigma = 0.5$ 1st (f) Walker2d $\sigma = 0.5$ 2nd (g) Walker2d $\sigma = 0.1$ 1st (h) Walker2d $\sigma = 0.1$ 2nd

(i) HalfCheetah $\sigma = 0.5$ 1st    (j) HalfCheetah $\sigma = 0.5$ 2nd    (k) HalfCheetah $\sigma = 0.1$ 1st    (l) HalfCheetah $\sigma = 0.1$ 2nd

Figure 3: Ablation study for data selection. The figure compares BAIL with the algorithm that simply chooses the state-action pairs with the highest returns (without using an upper envelope). The learning curves show that the upper envelope is critical components of BAIL.

## D.3   Ablation study using standard regression instead of an upper envelope

Figure 4 compares BAIL with the more naive scheme of using standard regression in place of an upper envelope. The learning curves show that the upper envelope is a critical component of BAIL.

(a) Training SAC,
Hopper

(b) Training SAC,
Walker2d

(c) Training SAC,
Ant

(d) Training SAC,
Humanoid

(e) Training DDPG
$\sigma = 0.5$, Hopper

(f) Training DDPG
$\sigma = 0.5$, Walker2d

(g) Training DDPG
$\sigma = 0.1$, Hopper

(h) Training DDPG
$\sigma = 0.1$, Walker2d

Figure 4: Ablation study using standard regression instead of an upper envelope. The figure compares BAIL with the more naive scheme of using standard regression in place of an upper envelope. The learning curves show that the upper envelope is a critical component of BAIL.

## E   Performance for execution batches

As discussed in the main body of the paper, we also performed experiments for execution batches. Once again, for a given algorithm, we use the same hyper-parameters for all environments and batches (training ane execution). We see from Table 2 that BC, MARWIL, BAIL, and BCQ have similar overall performance, with BC and MARWIL having the highest number of wins and also being slightly stronger in terms of average performance. MARWIL has one more win compared to BC, but slightly lower average performance. Comparing BAIL and BCQ, BAIL has a slightly stronger average performance score, and BCQ has a few more wins.

Table 2: Performance of Five Batch DRL Algorithms for 40 different execution datasets.

| ENVIRONMENT | BAIL | BCQ | BEAR | BC | MARWIL |
|---|---|---|---|---|---|
| M $\sigma = 0$ HOPPER B1 | **1026 ± 0** | 901 ± 132 | 4 ± 1 | **1026 ± 0** | **1026 ± 0** |
| M $\sigma = 0$ HOPPER B2 | 696 ± 233 | 805 ± 312 | 19 ± 23 | **977 ± 0** | **977 ± 1** |
| M $\sigma = 0$ WALKER B1 | 437 ± 20 | **525 ± 45** | 380 ± 194 | 444 ± 16 | 439 ± 17 |
| M $\sigma = 0$ WALKER B2 | 500 ± 12 | **554 ± 29** | 546 ± 28 | 489 ± 15 | **504 ± 4** |
| M $\sigma = 0$ HC B1 | 4057 ± 69 | 4255 ± 150 | **4470 ± 96** | 4032 ± 72 | 4073 ± 55 |
| M $\sigma = 0$ HC B2 | 4013 ± 12 | **4438 ± 25** | 4395 ± 31 | 3998 ± 4 | 3999 ± 6 |
| M $\sigma = 0$ ANT B1 | 753 ± 9 | **996 ± 52** | 734 ± 43 | 730 ± 7 | 732 ± 11 |
| M $\sigma = 0$ ANT B2 | 738 ± 4 | **994 ± 12** | 988 ± 30 | 708 ± 11 | 725 ± 7 |
| M $\sigma = 0$ HUMANOID B1 | 4313 ± 139 | 3108 ± 510 | 0 ± 0 | 4507 ± 481 | **4521 ± 156** |
| M $\sigma = 0$ HUMANOID B2 | **4053 ± 252** | 2906 ± 226 | 0 ± 0 | 3994 ± 530 | 3940 ± 165 |
| M $\sigma = \sigma(s)$ HOPPER B1 | 375 ± 52 | 881 ± 155 | 0 ± 0 | **1026 ± 0** | **1026 ± 0** |
| M $\sigma = \sigma(s)$ HOPPER B2 | 254 ± 102 | **961 ± 25** | 3 ± 7 | **977 ± 0** | **977 ± 0** |
| M $\sigma = \sigma(s)$ WALKER B1 | 384 ± 21 | 399 ± 21 | **507 ± 7** | 369 ± 10 | 359 ± 15 |
| M $\sigma = \sigma(s)$ WALKER B2 | 512 ± 24 | 517 ± 19 | 515 ± 30 | 527 ± 12 | **532 ± 5** |
| M $\sigma = \sigma(s)$ HC B1 | 4744 ± 19 | **5500 ± 12** | 5443 ± 21 | 4415 ± 25 | 4439 ± 59 |
| M $\sigma = \sigma(s)$ HC B2 | 4123 ± 19 | 4712 ± 40 | **4824 ± 51** | 3928 ± 18 | 3936 ± 18 |
| M $\sigma = \sigma(s)$ ANT B1 | 790 ± 9 | 1068 ± 12 | **1161 ± 32** | 775 ± 7 | 774 ± 15 |
| M $\sigma = \sigma(s)$ ANT B2 | 781 ± 6 | 1089 ± 29 | **1150 ± 18** | 768 ± 5 | 761 ± 6 |
| M $\sigma = \sigma(s)$ HUMANOID B1 | 1375 ± 387 | 489 ± 87 | 0 ± 0 | 1947 ± 901 | **1963 ± 264** |
| M $\sigma = \sigma(s)$ HUMANOID B2 | 1309 ± 372 | 816 ± 177 | 0 ± 0 | **3021 ± 1042** | 2976 ± 241 |
| O $\sigma = 0$ HOPPER B1 | **2602 ± 5** | 1976 ± 383 | 1904 ± 321 | 2594 ± 8 | **2603 ± 4** |
| O $\sigma = 0$ HOPPER B2 | 3046 ± 34 | 3014 ± 47 | 2202 ± 410 | **3071 ± 10** | 3050 ± 22 |
| O $\sigma = 0$ WALKER B1 | **2735 ± 26** | 2409 ± 235 | 877 ± 1077 | 2646 ± 133 | 2691 ± 121 |
| O $\sigma = 0$ WALKER B2 | **3019 ± 6** | **3019 ± 45** | 0 ± 0 | 3014 ± 5 | 3013 ± 5 |
| O $\sigma = 0$ HC B1 | 11265 ± 243 | 10405 ± 275 | 1755 ± 1142 | **11674 ± 90** | 11661 ± 49 |
| O $\sigma = 0$ HC B2 | 11360 ± 265 | 10792 ± 209 | 1139 ± 960 | **11797 ± 29** | 11691 ± 96 |
| O $\sigma = 0$ ANT B1 | 4901 ± 65 | 4646 ± 179 | 1756 ± 2151 | 4881 ± 74 | **4933 ± 74** |
| O $\sigma = 0$ ANT B2 | 4975 ± 108 | 4734 ± 100 | 0 ± 0 | **5041 ± 29** | 4974 ± 52 |
| O $\sigma = 0$ HUMANOID B1 | 4872 ± 895 | 4884 ± 641 | 0 ± 0 | 5462 ± 124 | **5503 ± 1** |
| O $\sigma = 0$ HUMANOID B2 | 5320 ± 125 | 5362 ± 54 | 0 ± 0 | **5413 ± 64** | **5413 ± 29** |
| O $\sigma = \sigma(s)$ HOPPER B1 | 2359 ± 153 | **2650 ± 99** | 1962 ± 300 | 1952 ± 85 | 2012 ± 101 |
| O $\sigma = \sigma(s)$ HOPPER B2 | 2035 ± 217 | 1678 ± 113 | 1461 ± 75 | 2063 ± 95 | **2092 ± 100** |
| O $\sigma = \sigma(s)$ WALKER B1 | 2834 ± 120 | **3386 ± 196** | 3278 ± 128 | 2024 ± 131 | 1987 ± 114 |
| O $\sigma = \sigma(s)$ WALKER B2 | 3200 ± 16 | **3375 ± 12** | 2100 ± 1715 | 3091 ± 15 | 3090 ± 10 |
| O $\sigma = \sigma(s)$ HC B1 | 10258 ± 1255 | 10928 ± 215 | 694 ± 651 | 11659 ± 75 | **11663 ± 44** |
| O $\sigma = \sigma(s)$ HC B2 | 10882 ± 634 | 11755 ± 97 | 1470 ± 1211 | **11871 ± 57** | 11819 ± 78 |
| O $\sigma = \sigma(s)$ ANT B1 | 4981 ± 91 | 4878 ± 117 | 3462 ± 1740 | **5000 ± 79** | 4992 ± 86 |
| O $\sigma = \sigma(s)$ ANT B2 | 5067 ± 83 | 5054 ± 157 | 0 ± 0 | 5079 ± 55 | **5124 ± 47** |
| O $\sigma = \sigma(s)$ HUMANOID B1 | 2129 ± 381 | 1715 ± 637 | 0 ± 0 | **3514 ± 1195** | 3180 ± 503 |
| O $\sigma = \sigma(s)$ HUMANOID B2 | 4328 ± 569 | 1970 ± 512 | 0 ± 0 | **4875 ± 885** | 4772 ± 272 |

# F   Learning Curves for all 62 Batches

## F.1   DDPG training batches

(a) Hopper, batch 1    (b) Hopper, batch 2    (c) Walker2d, batch 1    (d) Walker2d, batch 2

(e) HalfCheetah, batch 1    (f) HalfCheetah, batch 2

Figure 5: Performance of batch DRL algorithms on DDPG training batches with $\sigma = 0.5$. The policy networks for all algorithms are trained for 100 epochs except BAIL, which is trained for 50 epochs after training the upper envelope for 50 epochs.

(a) Hopper, batch 1    (b) Hopper, batch 2    (c) Walker2d, batch 1    (d) Walker2d, batch 2

(e) HalfCheetah, batch 1    (f) HalfCheetah, batch 2

Figure 6: Performance of batch DRL algorithms on DDPG training batches with $\sigma = 0.1$. The policy networks for all algorithms are trained for 100 epochs except BAIL, which is trained for 50 epochs after training the upper envelope for 50 epochs.

## F.2 SAC training batches

(a) Hopper, batch 1     (b) Hopper, batch 2     (c) Walker2d, batch 1     (d) Walker2d, batch 2

(e) HalfCheetah, batch 1     (f) HalfCheetah, batch 2     (g) Ant, batch 1     (h) Ant, batch 2

Figure 7: Performance of batch DRL algorithms on SAC training batches. The policy networks for all algorithms are trained for 100 epochs except BAIL, which is trained for 50 epochs after training the upper envelope for 50 epochs.

## F.3 SAC mediocre execution batches

(a) Hopper, batch 1     (b) Hopper, batch 2     (c) Walker2d, batch 1     (d) Walker2d, batch 2

(e) HalfCheetah, batch 1     (f) HalfCheetah, batch 2     (g) Ant, batch 1     (h) Ant, batch 2

Figure 8: Performance of batch DRL algorithms on SAC mediocre execution batches with $\sigma = 0$. The policy networks for all algorithms are trained for 100 epochs except BAIL, which is trained for 50 epochs after training the upper envelope for 50 epochs.

(a) Hopper, batch 1     (b) Hopper, batch 2     (c) Walker2d, batch 1     (d) Walker2d, batch 2

(e) HalfCheetah, batch 1     (f) HalfCheetah, batch 2     (g) Ant, batch 1     (h) Ant, batch 2

Figure 9: Performance of batch DRL algorithms on SAC mediocre execution batches with $\sigma = \sigma(s)$. The policy networks for all algorithms are trained for 100 epochs except BAIL, which is trained for 50 epochs after training the upper envelope for 50 epochs.

## F.4 SAC optimal execution batches

(a) Hopper, batch 1    (b) Hopper, batch 2    (c) Walker2d, batch 1    (d) Walker2d, batch 2

(e) HalfCheetah, batch 1    (f) HalfCheetah, batch 2    (g) Ant, batch 1    (h) Ant, batch 2

Figure 10: Performance of batch DRL algorithms on SAC optimal execution batches with $\sigma = 0$. The policy networks for all algorithms are trained for 100 epochs except BAIL, which is trained for 50 epochs after training the upper envelope for 50 epochs.

(a) Hopper, batch 1    (b) Hopper, batch 2    (c) Walker2d, batch 1    (d) Walker2d, batch 2

(e) HalfCheetah, batch 1    (f) HalfCheetah, batch 2    (g) Ant, batch 1    (h) Ant, batch 2

Figure 11: Performance of batch DRL algorithms on SAC optimal execution batches with $\sigma = \sigma(s)$. The policy networks for all algorithms are trained for 100 epochs except BAIL, which is trained for 50 epochs after training the upper envelope for 50 epochs.

## F.5 Learning curves for Humanoid

(a) training data, batch 1

(b) training data, batch 2

(c) mediocre $\sigma = \sigma(s)$, batch 1

(d) mediocre $\sigma = \sigma(s)$, batch 2

(e) mediocre $\sigma = 0$, batch 1

(f) mediocre $\sigma = 0$, batch 2

(g) optimal $\sigma = \sigma(s)$, batch 1

(h) optimal $\sigma = \sigma(s)$, batch 2

(i) optimal $\sigma = 0$, batch 1

(j) optimal $\sigma = 0$, batch 2

Figure 12: Performance of batch DRL algorithms with the Humanoid-v2 environment. All batches are obtained with SAC.

# G    Visualization of the Upper Envelopes

(a) Hopper $\sigma = 0.5$ 1st     (b) Hopper $\sigma = 0.5$ 2nd   (c) Walker2d $\sigma = 0.5$ 1st   (d) Walker2d $\sigma = 0.5$ 2nd

(e) HalfCheetah $\sigma = 0.5$ 1st     (f) HalfCheetah $\sigma = 0.5$ 2nd     (g) Hopper $\sigma = 0.1$ 1st     (h) Hopper $\sigma = 0.1$ 2nd

(i) Walker2d $\sigma = 0.1$ 1st   (j) Walker2d $\sigma = 0.1$ 2nd     (k) HalfCheetah $\sigma = 0.1$ 1st     (l) HalfCheetah $\sigma = 0.1$ 2nd

Figure 13: Typical Upper Envelopes for BAIL. For each figure, states are ordered from lowest $V(s_i)$ upper envelope value to highest. Thus the upper envelope curve is monotonically increasing. Each curve is trained with one million returns, shown with the orange dots. Note that the upper envelope lies above most data points but not all data points.

# H    Computing Infrastructure

Experiments are run on Intel Xeon Gold 6248 CPU nodes, each job runs on a single CPU with base frequency of 2.50GHZ.