[Reviews · NeurIPS 2020]

Review 1

Summary and Contributions: ---post author response--- Thank you for the response! The clarifications to the table have improved my understanding of the results. While I think that the results are strong, the discussion section is jumbled/unclear, and intuition of some of the design decisions are lacking and give an 'ad hoc' impression. Clarifications for this are adequately mentioned in the response, and I will increase my score to a 6 assuming the authors will add these clarifications to the final text, as well as make the experimental results section more more clear. ===== This work proposes a batch deep RL algorithm called BAIL. It essentially trains a policy using imitation learning with samples collected from state-action pairs whose (Monte Carlo) returns are from what the authors define as the upper envelope of the data. The upper envelope of the data are value function parameters such that,roughly, the sum of the difference between the value of each state $s_i$ and the sum of discounted returns from $s_i$ to the end of the episode is minimized. BAIL works for domains that have deterministic transition functions. An empirical evaluation was also conducted that compares the performance of BAIL against other batch deep RL algorithms. Against a particular set of domains, BAIL has higher performance than the baselines.

Strengths: The algorithm seems novel and is indeed straightforward and simple as the authors claim. Intuitively, it makes sense that selection of s-a pairs with high return from the data are used to perform imitation learning. The evaluation over five MuJoCo environments shows that BAIL in general is higher-performing. Some rows in Table 1 are bit misleading as not all instances bolded show the algorithm with the highest performance.

Weaknesses: There is little intuition or buildup to the two BAIL versions introduced by the paper---the reader has no sense of why selection of the best actions is performed in these two different ways. I think that a more thorough behavior analysis of BAIL would strengthen this work. For instance, since the algorithm seems dependent on the upper envelope of the data, the addition of experiments (synthetic or otherwise) with poor training examples (perhaps decreasing the quality of non-expert data) would be ideal to see how robust BAIL is to these instances. The execution batches experiments are towards this idea. Again, the table representing these results seems misleading. In 6 or 7 of the rows, BAIL does not have the best performance yet its element is high lighted. From the execution experiments provided, BAIL does not seem to perform competitively in this case. The question of "how close to expert does the training data have to be for BAIL to perform competitively" seems natural to ask for an algorithm that seems to mostly depend on high-quality state-action pairs. Perhaps I am misunderstanding the results, but it does not seem this is being answered well in the experimental section. BAIL is not mentioned to work in the case that environments transitions are non-deterministic.

Correctness: The empirical methodology seems fine. The authors provide reasons for most of their experiment decisions and (to me) nothing seems out-of-place.

Clarity: The paper is readable, but not very clear in some places, specifically explanation of the theorems and optimization problem for the upper envelope. I think that the related works section was nicely written. Also, it is not mentioned in the main body that there are proofs for the theorems (even though there are proofs in the appendix). This adds to the confusion of the paper but can be easily fixed by mentioning that the proof is in the appendix.

Relation to Prior Work: Yes.

Reproducibility: Yes

Additional Feedback: line 127>> the authors say Monte Carlo return G_i i sthe sum of discounted returns from state s_i to the end of the episode but then define G_i to actually be the sum of the discounted rewards.


Review 2

Summary and Contributions: This paper introduces BAIL, a new batch (also known as offline) RL algorithm. The algorithm using an interesting approach of estimating the upper envelope of data. The paper performs a large number of experiments comparing it to recent offline RL algorithms. Update: I've read your response, and my review stays the same. Solid work.

Strengths: I think the work is novel. The presentation is clear. The experimental evaluation is strong. And the results are convincing. I am fairly familiar with the literature in this area, and I think this work is solid.

Weaknesses: I think the results would be even more convincing if there were experiments are harder domains.

Correctness: The claims seem correct to me.

Clarity: The paper is well written.

Relation to Prior Work: The paper clearly discussed how it differs from previous contributions.

Reproducibility: Yes

Additional Feedback:


Review 3

Summary and Contributions: The authors introduce BAIL, a batch RL algorithm which estimates an upper bound on the value function and then uses this upper bound to filter out the best trajectories for imitation learning. The authors evaluate their approach on standard mujoco benchmark tasks and show that it compares favorably to other recent batch RL approaches as well as to behavioral cloning.

Strengths: The proposed method is simple to implement and straight-forward. The evaluation shows that it has the potential to be an easy way to achieve better performance without requiring additional environment interactions.

Weaknesses: The approach is at times ad-hoc and some choices could be better motivated. The evaluation isn’t entirely clear.

Correctness: The proposed approach is best seen as a heuristic. While it is intuitive that it can lead to good results when the given dataset is good, no guarantees are made that this is the case. The evaluation is relatively extensive and shows that such benefits can be observed.

Clarity: Most parts of the paper are relatively easy to follow and the method is easy to understand; however, the paper would benefit from an additional pass for spelling mistakes and stylistic faux-pas’s. It was unclear to me which results related to which method of collecting batches. Given the nature of the method, I was interested to see how well the agent would learn from sub-optimal training data which the execution batches in Section 5.1.2 might answer; however, I could not discern where the execution batches are being used.

Relation to Prior Work: The authors draw a connection to and evaluate against the recent body of work in BatchRL. The idea of using imitation learning on observed roll-outs that are better than the current estimate of the value has been explored before in Self-imitation Learning (Oh et al., 2018), albeit in a different context. The general idea of filtering an imitation learning training set to only include good episodes can also be found in the literature, and has for example been used in AlphaStar (Vinyals et al., 2019).

Reproducibility: Yes

Additional Feedback: In section 4.1, a lot of time is being spent on describing the notion of the upper envelope of the value function. This definition seems to be crucially dependent on the l2-regularization term, yet the authors do not use the l2-regularization and replace it with an early stopping criteria. It would be good to know why this is the case and how well the method would perform if the l2-regularization was actually being used. Due to the high-variance nature of the value estimate, it seems to me that the method might require a larger amount of high-quality training data when compared to other BatchRL approaches. As far as I can see, there is no evaluation with regards to that criterion. Theorem 4.1 seems gratuitous, the method is overall more of a heuristic and the theorem does not show anything surprising or lead to any guarantees for the algorithm.


Review 4

Summary and Contributions: The authors present a new batch RL algorithm based on the newly introduced notion of upper envelop for returns as well as the best action selection.

Strengths: * Solid experimental results as well as the range of the domains tested * Good theoretical analysis of the method

Weaknesses: * Computational analysis looks promising, but would be more convincing if authors demonstrated that the implementations of each of the baseline algorithms are efficient.

Correctness: Seems correct.

Clarity: Clearly written

Relation to Prior Work: Yes

Reproducibility: Yes

Additional Feedback: I would encourage authors to give more intuition to a reader on why BAIL performs so much better than other methods. **POST-REBUTTAL** After having read the authors response, I am inclined to keep my score.

[Author Response · NeurIPS 2020]

We would like to thank the reviewers for their insightful comments. As has been noted by all four reviewers, our paper presents a novel algorithm for batch reinforcement learning that is not only simpler and faster than the state-of-the-art algorithms, but also performs significantly better. Because the BCQ and BEAR code are publicly available, we were able to make a careful and comprehensive comparison of the performance of BAIL, BCQ, BEAR, MARWIL and naive behavioral cloning using the MuJoCo benchmark. For our experiments, we created non-expert training batches in a manner identical to what was done in the BCQ paper and included additional partially-trained training batches for the environments Ant and Humanoid using SAC. Our experiments showed that BAIL wins for 20 of the 22 batches, with overall performance 42% or more higher than the other algorithms. Moreover, BAIL is computationally 30-50 times faster than BCQ and BEAR. We provide robust anonymized code for reproducibility. We will also make our datasets publicly available for future benchmarking.

BAIL learns a value function by training a neural network to obtain the "upper envelope of the data". To the best of our knowledge, the notion of the upper envelope of a dataset is novel, and can possibly be applied to other RL problems in the future.

**Response to Reviewer 2:** Thank you for your positive feedback with an accept recommendation. In terms of testing on alternative domains, we are currently focused on MuJoCo, where Ant and Humanoid are the most challenging environments. But we will consider testing on alternative benchmarks in future work.

**Response to Reviewer 4:** Thank you for your positive feedback with an accept recommendation. We will add to the paper more intuition on why BAIL performs better than the other methods. Intuitively, BAIL performs better than BCQ and BEAR because BCQ and BEAR rely on carefully tuned policy constraints to prevent the use of out-of-distribution actions. A loose constraint can cause extrapolation error to accumulate quickly, and a tight constraint will prevent the policy from choosing some of the good actions. BAIL, however, identifies and imitates the highest-performing actions in the dataset, thus avoiding the need to carefully tune such a constraint. In terms of the computation, for BCQ and BEAR, we used the authors' implementations. These Q-learning based algorithms typically have more, larger networks, and sample multiple candidate actions for each update. Thus it is not surprising that BAIL can be 30-50 times faster.

**Response to Reviewer 3:** Thank you for your mostly positive review, and for pointing out the papers (Oh et al, 2018) and (Vinyals et al, 2019), which we will add to the related work section. In our view, all DRL algorithms are heuristics, and performance guarantees for schemes using neural-network function-approximators are rare. We remark the experimental results presented in the main body indeed use "sub-optimal training data," since those batches were obtained from training data well before optimal performance is achieved (and in many cases using sub-optimal algorithms for training). We will make this more clear in the revision. The results for the execution batches are summarized in the main body and presented in detail in the appendix.

We decided to use $L_2$ regularization in the definition of the upper-envelope since it leads to a clean definition and theory. In our computational experiments, we found $L_2$ regularization and early-stopping regularization to give similar performance, with early stopping being faster. So we chose to use early stopping in our experiments. We note that there is a well-known theory showing, under some conditions, the equivalence of early stopping to $L_2$ regularization (see the Deep Learning book by Goodfellow et al., Section 7.8).

**Response to Reviewer 1:** Thank you for the feedback. It seems you are very concerned about our table results. We would like to point out that they are definitely not "misleading". In fact, the table is presented in this manner to make fair comparisons, as clearly explained in the paper. On page 8, line 277, we explain: "For each batch, all the algorithms that are within 10% of the highest average value are considered winners and are indicated in bold." So it is possible for multiple algorithms to be in bold in a table row. Indeed, when two algorithms have very similar scores, this should be considered a tie, since their rankings can change easily when using a different random seed. To complement the tables, on page 8, line 283, we also explain how we computed the average improvement ratio, which shows that BAIL performs 42% better than BCQ, and 101% better than BC.

You ask "how close to expert does the training data have to be for BAIL to perform competitively?" This question is answered starting at page 2, line 39, where we explained that we generated a large variety of batches, including far-from-expert datasets with different random noise levels and with different learning agents. Our batch generation procedure is also consistent with the procedures in prior work. Thus Table 1 answers your question by showing that BAIL performs well on all the training datasets with various noise levels.

For the execution batches, we used the same hyper-parameters for the training set, and the results show that BAIL is better than BEAR, similar to BCQ, and slightly weaker than naive behavioral cloning. If we change the hyper-parameters we will get at least the same performance as behavioral cloning (which is the strongest in the execution batches). Thus, in summary, BAIL is the clear winner for the training batches, and BAIL, BCQ, and naive behavioral cloning are equally good for the execution batches. For the other minor points you suggested, thank you for pointing them out, and we will modify them in our revision.

[Meta-Review · NeurIPS 2020]

The authors agreed that the paper makes good contributions to batch RL, and the rebuttal has been very helpful. Some concerns around the empirical evaluation remain, but the paper makes a good contribution. Please make sure that the revised version of the paper actually reflects the rebuttal and reviewer recommendations.